# Clinical Evaluation of Magnesium Alloy Osteosynthesis in the Mandibular Head

**DOI:** 10.3390/ma15030711

**Published:** 2022-01-18

**Authors:** Marcin Kozakiewicz, Izabela Gabryelczak, Bartosz Bielecki-Kowalski

**Affiliations:** Department of Maxillofacial Surgery, Medical University of Lodz, 113 Żeromskiego Str., 90-549 Lodz, Poland; gabryelczakizabela@gmail.com (I.G.); bartosz.bielecki-kowalski@umed.lodz.pl (B.B.-K.)

**Keywords:** magnesium, mandible head, mandible condyle, mandible fracture, condylar head fracture, fracture treatment, fixing material, osteosynthesis, open rigid internal fixation, surgical treatment

## Abstract

Titanium alloys are used in skeletal surgery. However, once bone union is complete, such fixation material becomes unnecessary or even harmful. Resorbable magnesium materials have been available for several years (WE43 alloy). The aim of this study was to clinically compare magnesium versus titanium open reduction and rigid fixations in mandible condylar heads. Ten patients were treated for fractures of the mandibular head with magnesium headless compression screws (2.3 mm in diameter), and 11 patients were included as a reference group with titanium screws (1.8 mm in diameter) with similar construction. The fixation characteristics (delay, time, and number of screws), distant anatomical results (mandibular ramus height loss, monthly loss rate, and relative loss of reconstructed ramus height), basic functional data (mandibular movements, facial nerve function, and cutaneous perception) and the influence of the effects of the injury (fracture type, fragmentation, occlusion, additional fractures, and associated diseases) on the outcome were evaluated. The long-term results of treatment were evaluated after 18 months. Treatment results similar to those of traditional titanium fixation were found with magnesium screws. Conclusions: Resorbable metal screws can be a favored option for osteosynthesis because surgical reentry can be avoided. These materials provide proper and stable treatment results.

## 1. Introduction

Titanium alloys are used in skeletal surgery, in both children [1,2,3] and adults [4,5,6,7]. Long screw fixation through the lateral fragment end of the ascending ramus (i.e., distal fragment) has been a well-known procedure for osteosynthesis of the condylar head for approximately the last 30 years [8]. Unfortunately, for most of that time, titanium alloy screws were used. This is partly due to the need for low-profile screws in the mandibular head [9]. Very narrow compression screws, either cannulated [10] or solid [11], are used in the mandibular head. This solves many clinical issues and prevents iatrogenic destruction of mandibular head fragments by screws that are too thick. However, once bone union is complete, this fixation material becomes unnecessary [12] or even harmful [13,14]. Titanium screws in the mandible cause a number of functional abnormalities at the cellular level: They increase local proinflammatory cytokine levels, enhance free radical generation in the periosteum covering implants, and induce apoptosis in the mandible periosteum [14], oxidative and nitrosative stress, or disturbances in mitochondrial function [13]. Titanium alloys are not resorbable in the human body. For this reason, resorbable materials [15,16,17] are sought for mandibular head surgery.

Consequently, it is recommended that nonresorbable metal components previously used for fixation be routinely removed to avoid long-term interference with periarticular soft tissues [14]. In particular (where the condylar head is fixed), direct contact of fixation material with the capsule and disc attachments in the lateral condylar pole zone leads to cicatrization, with negative effects on disk and condylar mobility [18]. Neff and Kolk’s strong belief in the need to remove osteosynthesis material stems from the known clinical effects of leaving nonresorbable fixative material in any part of the skeleton (some are not even as sensitive as the mandibular head): stress shielding, metallosis, migration, radiation and X-ray effects, palpability, reinjury, thermal sensitivity, loose hardware, perforation exposure, or infection. However, metal removal remains a debated topic [19]. As removal of the fixation material is considered particularly risky in mandibular condylar process fractures, especially in fractures of the head itself (leading to possible facial nerve palsy), most studies to date do not consider postoperative metal removal [20,21,22,23]. The use of a retroauricular approach instead of a preauricular approach is only a partial solution to the problem [24] because type C fracture fixation is easier to perform with a preauricular approach. In the case of resorbable polymers, the acidic degradation process will affect intra- and periarticular soft tissues, which may cause additional scarring that significantly disturbs function [25], as previously described for PLLA-PGA-based resorbable screws [26,27]. It is also worth noting that Young’s modulus and other main mechanical properties of magnesium can be close to that of compact bone and are also much higher those that of PLLA-PGA [28,29].

For these reasons, materials composed of resorbable magnesium alloys appear to be another candidate for testing in mandibular head fracture osteosynthesis. The first attempts have already been made [30,31]. The aim of this study was to compare magnesium versus titanium open reduction and rigid fixation (ORIF) in the head of the mandibular condyle.

## 2. Materials and Methods

The study was authorized by a bioethics committee (corresponding ethical approval code: RNN 227/19/KE). Twenty-one patients affected by fracture of the mandibular head (10 in the test group and 11 in the reference group) were included in this study: 3 females and 18 males. Inclusion criteria: recent trauma, condylar head fracture type B or C according to Neff’s classification [27], surgical treatment, preauricular approach, and complete radiological documentation. Patients that underwent operations on Tuesdays and Thursdays received titanium fixation, while patients that underwent operations on Wednesdays and Fridays received magnesium fixation (the choice of fixation material was decided on the day the patient arrived at the hospital). Exclusion criteria: type A head fractures [27], failure of the patient to report for follow-up examinations, close (i.e., conservative) treatment, and old fractures (i.e., more than 4 weeks old). Dedicated mandibular head fracture fixation compressive screws (ChM, Juchnowiec Kościelny, Poland) were used in this study [31]. An additional 12 patients underwent surgery, but they only reached the 6- to 9-month follow-up period.

Titanium headless compression screws 1.8 mm in length and 14, 16, or 18 mm in length were used in this study (reference group: 11 patients). The titanium alloy was Ti_6_Al_7_Nb. Magnesium 2.3 mm headless compression screws [11] of the same length as the titanium screws were used (test group: 10 patients). The magnesium alloy was MgYREZr (i.e., WE43 MEO Mg alloy WE43 MEO with a nominal composition (in wt%) of 1.4–4.2% Y, 2.5–3.5% Nd, <1% (Al, Fe, Cu, Ni, Mn, Zn, Zr) and balance Mg). No cannulated screws were used in the study, and only solid screws were used. The primary clinical rationale for using a specific number of screws for the patient was the need to achieve primary stabilization of bone fragments during ORIF while maintaining the integrity of the mandibular head. All patients were operated on under general anesthesia with nasotracheal intubation with preauricular access. Data on the anatomy of the mandibular ascending ramus both before and at both periods of the postoperative study (immediate and 18 months post-operational) were obtained by analysis of spiral multislice computed tomography (Figure 1) and classified in accordance with AOCMF Classification System [32].

Moreover, clinical data were collected. All patients were followed up in the outpatient clinic after hospital treatment (in the same way as in other patients from the department). Thus, the postoperative evaluation was carried out systematically (every three months) in a standardized manner (using an evaluation sheet). The following variables were measured postoperatively in the study groups:Evaluation of the height of the mandibular ramus: To assess bone resorption in the mandibular head and loss of mandibular ramus height, multislice radiographs (computed tomography) were performed immediately after surgery and at the 18-month postoperative follow-up. Loss of vertical bone height was determined by the length from the angle of the mandible to the highest point of the articular surface on lateral views. The result of segmentation of the mandibular bone after removal of the images of the other bones of the skull and cervical spine and the technique of measuring the height of the mandibular ramus is shown in Figure 2. The loss of the mandibular ramus height was expressed in three quantities in this study: ramus height loss, ramus loss per month, and relative ramus loss (to intact side). These describe the same phenomenon but in three ways.Assessments of facial nerve disfunction were conducted at 3 days, 3 months, 6 months and 18 months after surgery based on House and Brackmann’s classification of facial nerve dysfunction severity [33]. This approach allows for the degree of damage to be assessed and the healing process to be traced. There are six degrees of nerve damage. Possible scores are 1 to 6 (higher scores correspond to extended damage). One person examined all groups included in this study in a blinded fashion.The Helkimo dysfunction index [34] was analyzed 3 days after the operation (early) and one year after the operation (late). The clinical index of temporomandibular joint dysfunction includes the amplitude of mandibular movements, disordered joint function, muscle pain during intraoral examination, temporomandibular joint pain during extraoral examination, and pain during jaw movements. A Helkimo index score of 0 points implies good functional results, scores between 1 and 4 points show insignificant dysfunction, and scores between 5 and 9 demonstrate average disfunction outcomes. A sum score of 10 points marks a poor functional outcome.

In the distal fragment (i.e., ramus fragment), the distance from the upper border of the screw socket to the fracture line was calculated (Figure 3). This appears to be critical for the stability of the fixation, as the length of the screw core in the proximal fragment (i.e., condylar fragment) is usually much longer than its part in the distal fragment. This may have been a weak point in the osteosynthesis.

The angle at which the screws were inserted into the external bone surface of the distal fragment (ramus) was measured. A 90-degree angle indicates perpendicular insertion of the screw. Angles greater than 90 degrees indicate an upward oblique screw insertion. It has also been noted that the screws can pass through the bone. In such cases, how many millimeters the screw protrudes on the internal side of the ramus was determined.

In the CT image taken 18 months after treatment, the fracture fissure site was marked as the region of interest, and the mean value in Hounsfield units (HU) was determined (postfracture site). A similar procedure was followed with the image of the intact trabecular substance in the mandibular head (control site). The optical density at the fracture site after the remodeling period was thereby assessed. Texture in the CT images taken 18 months after surgery was analyzed using MaZda 4.6 software developed by the University of Technology in Lodz Poland [35] to check the quality of restored bone structure [36]. Data were divided into two groups: postfracture data from the site where the fracture line ran and control data from intact normal cancellous bone of the mandible head. These regions of interest (ROIs) were normalized (*μ* ± 3*σ*) to share the same average (*μ*) and standard deviation (*σ*) of optical density within the ROI. Selected image texture features (sum of squares) in ROIs were calculated for the postfracture site and for control bone:(1)$$SumOfSqrs=\sum_{i=1}^{N_{g}}\sum_{j=1}^{N_{g}}{\left(i-\mu_{x}\right)}^{2}\;p\left(i,j\right)$$
where Σ is the sum; *μ_x_* is the mean of the row sums of the co-occurrence matrix; *N_g_* is the number of distinct gray levels in the quantized CT image, where *N_g_* is equal to 2*^k^*, and *k* is the number of bits per pixel; *p*(*i*,*j*) is the number of times there is a run of length; and *j* with gray level *i* is the optical density at a distance of two pixels.

Gas formation was observed in the group of patients treated with magnesium screws. The volume of gas produced was measured according to the following protocol: CT images in Digital Imaging and Communications in Medicine format were segmented and transformed into a 1-bit three-dimensional model based on individual histogram analysis according to the Baillard and Barillot protocol [37] using Mimics 17.0 software (Materialise, Leuven, Belgium). Values corresponding to formed gas were selected, and only gas formation within the mandibular head bone was selected. The volumes of the selected gas bubbles in the images were measured using the stability function in freeware Meshmixer software (Autodesk, San Rafael, CA, USA).

Statistical analysis included feature distribution comparison, mean or median comparison, analysis of regression for quantitative variables, and one-way analysis of variance assessing the impact of a qualitative variable on a quantitative variable (e.g., fixative material on treatment outcome). The relationship between a qualitative feature (e.g., fracture type, presence of internal disease in the patient, dislocation, occlusion) and the fixation material used was assessed using the χ^2^ test. Detected relationships were assumed to be statistically significant when *p* < 0.05. Factor analysis was applied as a statistical method to extract one unobserved variable called a factor from the clinical data of patients, which helps to find data influencing mandibular ramus loss in the postoperative period. This involves statistically extracting the broadest possible description of variability from the clinical features and infusing it into one newly referenced variable (i.e., factor) while retaining this original information. Then, one number can describe the influence of three features on the final height of the mandibular ramus. A factor that had eigenvalues greater than or equal to 1.0 was sought. Three peri-operational variables were chosen: mandibular ramus shortening produced by bone fragment overlapping after injury, mandibular head fragmentation (i.e., number of radiologically observed fragments except the distal ramus fragment connected to the mandible ramus) and Helkimo disfunction index registered during early posttreatment. Statgraphics Centurion version 18.1.12 (StatPoint Technologies Inc., Warrenton, VA, USA) was used for statistical analyses.

## 3. Results

Assessing using CT whether union of the mandibular head fragments had occurred, it was found that the mean optical density of the fracture site 18 months after healing (Figure 4, Table 1) reached a value slightly higher than that of cancellous bone in the test group with 448 ± 122 HU (cancellous control bone 419 ± 81 HU; *p* = 0.539) and in the reference group with 408 ± 122 HU (cancellous control bone 384 ± 85 HU; *p* = 0.599). There was no difference between these mean bone densities in the test group and reference group (*p* = 0.453).

Digital bone texture analysis at the fracture site shows that after 18 months of healing, the sum of the squares of the optical densities at a distance of two pixels is the same as in the test group and in the reference group. Furthermore, the bone union quality measured in this way (magnesium fixation vs. cancellous control bone *p* < 0.05; reference fixation vs. cancellous control bone *p* < 0.05) is distinguishably denser bone than the cancellous control bone (Table 1 and Figure 4).

The groups compared are presented in Table 2 and Figure 5 and Figure 6. There were statistically significant differences (*p* < 0.05) found for the number of compression screws used (reference vs. test = 1.9 vs. 2.4), length of screws used (reference vs. test = 16 vs. 14 mm), and final (i.e., after 18 months) mouth opening (reference vs. test = 39 vs. 43 mm). There was no relationship between mandible height loss and screw material, screw number, or screw length (Figure 5).

Gas formation was observed in the test group. The volume of gas accumulated within bone 3 months after magnesium osteosynthesis was significantly reduced during the following 3 months from 52 ± 39 to 11 ± 18 mm^3^ (*p* < 0.05). The screws did not fully resorb during the observation period.

An attempt was made to combine several clinical features identifiable early during treatment into a single factor. It was found that the three selected features can form a single factor (called “mandible head status”) because the eigenvalue was 1.45. The mandible head status can be calculated using the following equation:Mandible Head Status =0.0715823 × Ramus Shortening + 0.922357 × Head Fragmentation + 0.62699 × Helkimo Disfunction Index Early(2)

The main contribution to mandible head status comes from the number of fragments within the mandibular head (over 92% of the source variation is drawn to the factor). In contrast, the amount of posttraumatic shortening/overlapping of the fragments itself has a small contribution (only 7% of the source variability). Higher values of this factor indicate greater pathomorphological and pathophysiological disturbances after mandibular head injury. Clinically, values from approximately 1 to approximately 30 can be observed. The mandible head status (Figure 7) was 5.50 ± 2.52 in the test group and 3.71 ± 2.10 in the reference group (ANOVA *p* = 0.050, Kolmogorov–Smirnov test *p* = 0.120 to compare distribution and Mann–Whitney (Wilcoxon) W-test to compare medians *p* = 0.113 confirmed similarity of both groups).

The gas (hydrogen) synthesis observed in the human body, which clearly occurs at the initial stage of magnesium resorption, should also be mentioned. This gas accumulates in the first 2 to 4 months and then gradually disappears (Figure 8). No negative effects on bone healing were observed that could be related to this production and the presence of gas, except for the detected relationship of greater mandibular abduction 6 months after treatment (higher MIO) in patients with more gas observed in the third postoperative month (correlation coefficient, CC = 0.75; R^2^ = 57%; *p* < 0.05).

The distance from the fracture gap edge of the highest screw in the distal fragment (superior screw) was 3.8 ± 1.9 mm (range: 1.4–9.7 mm; 21 cases), that of the inferior screw was 6.8 ± 3.0 mm (range: 1.8–14.0 mm; 20 cases), and that of the third screw located anteriorly to the other two screws was 5.2 ± 0.8 mm (range: 4.5–6.3 mm; four cases). Due to the small number of cases stabilized with three screws, an average was calculated for the two screws (the superior position and the inferior one), and a relatively weak relationship with the monthly loss of mandibular ramus height was noted (CC = 0.44; R^2^ = 19%; *p* < 0.05).

In the case of the test group, the distance of the superior screw from the fracture line was 3.1 ± 1.7 mm (vs. reference group: 4.5 ± 2.0 mm; no significant difference), 5.5 ± 3.5 mm from the inferior screw (vs. reference group: 8.2 ± 1.8 mm; *p* < 0.05), and 5.2 ± 0.8 mm from the anterior screw (no titanium fixations performed with three screws). In the magnesium fixation group, the distances of the superior, inferior, and anterior screws were not related to the loss of height of the mandible ramus. There was also no relation between the average distance of the superior and inferior screws together or the average distance calculated for all three screws together from the fracture gap. Ramus loss per month was not related to the distance of the superior or anterior screw from the fracture line but was moderately strongly related to the position of the inferior screw (CC = 0.76; R^2^ = 58%; *p* < 0.05), the average distance calculated for the two screws, i.e., superior and inferior (CC = 0.70; R^2^ = 49%; *p* < 0.05); and all three screws together (CC = 70; R^2^ = 48%; *p* < 0.05). In reference osteosynthesis, neither ramus height loss nor ramus loss per month were related to any screw position (Figure 9).

Evaluation of screw angulation (to the lateral bone surface) showed that the angle of insertion of the superior or inferior screw was not related to the amount of loss in ramus height (neither was the amount of protrusion of the tip screw on the medial side of the mandible for the superior and inferior screws). Due to the small number of cases with a third screw, no statistical study of this fixation point was conducted. Similarly, no relationship was noted between angulation or protrusion of the screw and monthly loss of height of the mandibular ramus or between angulation and relative loss of ramus (i.e., observed difference between the operated side to the intact side).

The average monthly loss of height was 0.6 ± 0.8 mm in whole patient group, and the relation loss at 18 months after surgery was 6.5 ± 4.5%. The average loss in the presented group of patients was 4.2 ± 3.1 mm. The percentage of mandibular ramus loss during the follow-up period was examined in relation to the original reconstructed mandibular ramus height (relative decrease in ramus). It was noted that it was not related to patient age; mandibular head fragments; MIO before treatment; ipsilateral movement before treatment; posttraumatic shortening of the mandibular ramus; delay to treatment; number of screws used for mandibular head fixation; length of screws used, MIO and laterotrusions immediately after surgery, 3 months after surgery, or 6 months after surgery; MIO 18 months after surgery; ipsilateral movement 18 months after surgery and facial nerve status immediately after surgery; or mandible head status. While it was related to early Helkimo disfunction index measurement (CC = −0.50; R^2^ = 25%; *p* < 0.05), it was even more strongly associated with late Helkimo disfunction index measurement (CC = −0.67; R^2^ = 45%; *p* < 0.001), contralateral movement before treatment (CC = −0.97; R^2^ = 94%; *p* < 0.01), facial nerve status 18 months after treatment (CC = −0.72; R^2^ = 51%; *p* < 0.001; although only one patient had score 2 and one patient had score 3, and the rest achieved score 1, i.e., full recovery), range of contralateral mandibular movement 18 months after treatment (CC = 0.57; R^2^ = 33%; *p* < 0.05), and understandably (as these are similar measures) with ramus height loss (CC = 0.99; R^2^ = 99%; *p* < 0.001) and ramus loss per month (CC = 0.93; R^2^ = 86%; *p* < 0.001).

## 4. Discussion

There are two alternatives for titanium alloy osteosynthesis: polymeric fixation systems (especially for children) [38] and resorbable metals [39]. Testing of a screw model comparing titanium vs. polymeric screws and polymeric vs. magnesium screws indicated that magnesium screws are more resilient than polymeric screws [9]. Currently, magnesium materials seem more attractive than polymers in condylar head fixation. There are clinical limitations for using low-profile screws in the mandibular head due to the material features of WE43 MEO magnesium alloy. The pull-out force being higher for the Mg screw than for the Ti screw [31] results from the screw design (higher shaft diameter and thread depth) and not the material property differences. It is known [11] that torque is directly proportional to screw diameter. Regardless, the polymer screw has the lowest axial pull-out force. Although Mg screws are thicker than Ti screws, the torsional properties of narrower Ti screws are better [31]. This is due to the high Young’s modulus of titanium alloys. Therefore, a titanium screw should be selected if a screw with high resilience is needed and Mg for more standard osteosynthesis, and magnesium requires multiscrew fixations [31]. Among the polymer, titanium, and magnesium screws, the metal biodegradable magnesium screw seems to be the most suitable material for mandibular head fixation, considering the condition of the fragile screwdriver socket [9,31]. The stability of bone fragments fixed by magnesium screws weakens with time after osteosynthesis (in two months it is as weak as a polymeric screw). The decrease in the pull-out force that occurs with the progression of screw resorption may cause early bone healing to be vulnerable to displacement, i.e., loss of mandible ramus height. For this reason, it seems clinically reasonable to investigate magnesium osteosynthesis and to recommend using more screws (Figure 1) and/or combining thinner screws of 1.8 mm with thicker screws, i.e., 2.3 or 3.2 mm [40].

Based on optical density assessment, it was found in this study that there was bone union at the mandibular head fracture site. The density measured in HU was even slightly higher at the fracture site than in the surrounding cancellous bone. Thus, the recreation of proper quality bone (cancellous) was achieved in the patients presented here. This observation was also confirmed by the analysis of the co-occurrence matrix, i.e., sum of squares of optical density of pixels in CT scans. The evaluation of the bone structure at the fracture site 18 months after fixation revealed a slightly elevated density with features of diffuse opaque structures immersed in normal opaque bone. Thus, typically, the fracture site changes its structure to slightly denser (but this is not statistically detectable). Bone consolidation at the osteosynthesis site was achieved in the magnesium group and in the reference group.

The location of the magnesium screw in the distal fragment (lower ramus fragment) appears to be important for maintaining the height of the reconstructed mandibular ramus. This is the first such study and requires follow-up, but at this stage, it can be concluded that excessively lower placement of the inferior screw is associated with progressive loss of mandibular ramus height resulting from mandibular head tilt over several months after the surgical procedure (Figure 9). It is also worth emphasizing that the superior screw positioned at an average distance of 3.8 mm from the edge of the fracture line is not related to the risk of reducing the height of the mandible ramus (the headless screw is sufficiently fixed in the distal fragment). Evaluation of the screw insertion angle and the size of the screw medial protrusion revealed that the large insertion angles of the fixation material and the through-and-through puncture did not affect the loss of height of the mandibular ramus 18 months after surgical treatment.

In vitro findings [41] indicated concentration-dependent effects of magnesium resorption (corrosion is the same phenomenon, but is usually described in vitro or outside of the organism, and it is called “resorption” if it is observed in the body) products on the cellular activity of osteoclast progenitor cells. Magnesium significantly altered the metabolism and proliferation of these cells. While cells tolerated magnesium concentrations higher than the physiological range (16 mM), concentrations below 10 mM were beneficial for cell growth. New studies are currently awaited to deliver biochemical, molecular, and epidemiological data or descriptions of the side effects of magnesium in humans, which would be important and interesting findings.

Current evidence from preclinical experiments suggests that the application of magnesium implants (over titanium screws) is beneficial to promote fracture healing. However, considerable heterogeneity existed among studies regarding animal species, implant preparation, surgery, and evaluation techniques. The literature also revealed the limitations in methodological quality and risk of bias among in vivo surgical studies to be improved via detailed planning and reporting of randomization. The findings indicated that there is still a lack of a standardized reference model to develop magnesium screws for fracture fixations. Nevertheless, the technical details extracted from published articles may function as building blocks for comprehensive study design and standardized protocols. This evidence may provide useful information on the selection of clinically relevant models, design of implants, and evaluation techniques for planning and conducting preclinical research with a human translational perspective [42].

This resulted in the final decision on which screw to use clinically in this study. However, it is not the only option, especially in multifracture fractures. A new approach from the Kiev team [5,6] with custom-made plates for proximal fragment reduction is a very interesting solution to a serious clinical issue. Consideration could be given to manufacturing these customized plates with magnesium alloy.

The mechanism of resorption of magnesium alloys in the human body is not fully understood [43], but it seems to be based on the same reactions that cause corrosion in aqueous environments [44]:Mg → Mg^2+^ + 2e^−^(3)
2H_2_O + 2e^−^ → H_2_ + 2OH^−^(4)
Mg^2+^ + 2OH^−^ → Mg(OH)_2_(5)

Certainly, the fates of other chemical elements present in the alloy (e.g., Y, Nd, or Zr) are different from that of magnesium [45,46] in the skeleton. Given the multitude of known magnesium alloys, it is likely that the distant reactions as well as the detailed effects on the surrounding bone microenvironment will vary. Therefore, it seems that a large chapter in medicine is opening concerning the study of magnesium fixation in traumatology.

The phenomenon of gas (H_2_) accumulation during the initial resorption phase of a magnesium screw has been reported [47]. For all clinicians using titanium screws thus far, it is surprising and worrying. It seems that separation of the fragments of cancellous bone by gas would disrupt the union of the fragments. However, this is not the case. There are no reports of clinical complications caused by accumulating gas (this study also did not find a relationship between gas and a decrease in ramus height). However, it is important to follow this phenomenon in the literature and its effect on bone healing. It would be interesting to see the results of studies on osteoblast/osteoclast activity in a hydrogen atmosphere. Such studies are not currently being conducted. Gas accumulation is one of the differences observed in fixations with magnesium material compared to titanium.

In our study, the amount of gas released during the resorption process was evaluated using segmentation based on CT images. At this point, it should be noted that titanium (and even more so chromium-cobalt) fixations significantly degrade the CT image and reduce its diagnostic power. It is noteworthy that magnesium, as a lighter metal alloy, causes fewer artifacts in CT images [48]. This allows for a more accurate analysis of CT images, especially subtle changes around the screw heads such as resorption (which often determines their removal). Smaller artifacts also facilitate segmentation and 3D modeling.

The financial aspect is also important. Magnesium screws are more expensive, making fixation more costly [47]. However, as noted by Böstman in his paper summarizing the costs of the initial surgery as well as the costs of a second operation with the removal of titanium screws, the total cost of treatment in these cases is in favor of bioresorbable materials [49].

It should be noted that the groups compared here were similar in many of the characteristics assessed (age, preoperative mouth opening, movement ipsilaterally to the fracture or movement contralaterally from the fracture, preoperative observed mandibular ramus shortening, mandibular head fragmentation, early surgical treatment initiation, and duration of surgery). In addition, many of the characteristics studied were similar in both groups postoperatively (magnitude of mandibular ramus height loss after bone remodeling period, rate of ramus loss per month, restored mandibular ramus height, relative loss in mandibular ramus, Helkimo dysfunction index both during early and long-term posttreatment, immediate posttreatment mouth opening, movement ipsilaterally to the fracture and movement contralaterally from the fracture, facial nerve disfunction immediately after surgery and at 18 months after treatment, movement of the mandible toward the treated fracture at 18 months, and movement toward the side opposite the treated fracture). Therefore, the authors preliminarily rated magnesium fixation of the mandible head as being as effective as titanium fixation performed with a dedicated headless compression screw.

The group of patients treated with magnesium alloy screws required, on average, a higher number of screws per fixation than the reference group due to the reasons cited above [31,40]. On the other hand, shorter magnesium screws were used than titanium screws. Shorter screws are more difficult to damage by twisting. Due to the good results of mouth opening (MIO) and similarity in terms of mandibular ramus loss, it seems that the clinical use of magnesium screws can be considered fruitful, and other aspects of magnesium use, such as gas formation, are not clinically relevant.

The overall results obtained, i.e., joint results for test and reference groups, should be regarded in the rather new field of maxillofacial surgery, i.e., fracture fixation of mandibular heads [34]. After surgical treatment of the mandibular head fracture, good functional results are obtained at 18 months, despite the observed reduction in mandibular ramus height. Clinical examination shows an average monthly loss of height of 0.6 ± 0.8 mm and a loss in relation to the original surgically achieved height at 18 months after surgery of 6.5 ± 4.5%, giving an average loss in the presented group of patients of 4.2 ± 3.1 mm. It is worth noting that the presented previous results of surgical treatment of mandibular head fractures [7] in groups of 28 and 35 patients describe only changes in bone height in the mandibular head part. However, it seems that bone remodeling also occurs within the entire ramus and not only in the mandibular head from the screw to the articular surface. Furthermore, by examining the entire height of the mandibular ramus, it is possible to obtain results comparable to those of closed/conservative treatment (where there is no screw in the mandibular head and no way to measure atrophy from the screw to the articular surface).

The results of conservative treatment are inferior to those of surgery, i.e., an average loss of mandibular ramus height of 4.5–5.4 mm in closed/conservative treatment [50]. Said publication uses a slightly different method to assess ramus loss than this study, but in any case, one of the main differences between the two treatment methods was the mean loss of vertical ramus height (compared with the height on the opposite, intact side). The decrease in vertical ramus height was 4.50 ± 1.56 mm in type B fractures and 5.4 ± 3.21 mm in type C fractures. In addition to the decrease in vertical height of the mandibular head bone [7], a loss of ramus height was also found in type B and C fractures. In type B and C fractures, after conservative treatment, the proximal fragment (the small fragment), including the articular disc, remained permanently dislocated in an anteromedial direction, which was associated with a change in the direction of pull of the lateral pterygoid muscle. In contrast, in osteosynthesis-treated patients, condylar height decreased by only 0.50 ± 0.87 mm for type B fractures and 0.77 ± 0.88 mm for type C fractures. Furthermore, in contrast with conservative results, surgical treatment results in full anatomically correct repositioning of the proximal fragment and the disc.

It seems that the advantage of surgical treatment results, which are superior to those of closed treatment [50], is the relation observed in this study between the good results of contralateral mandibular movement from the fracture side even with and higher relative reductions in ramus height. This can be explained by the occurrence of a loss of ramus height but with the preservation of an anatomically permanent attachment site of the lateral pterygoid muscle and stable restored disc position. In contrast, in conservative treatment, the attachment of this muscle migrates with the articular disc and the mandibular head fragment anteriorly and medially, significantly disrupting this laterotrusion. In the opinion of the authors of this study, restoring the normal range of motion of the laterotrusion is one of the primary goals of treatment of mandibular head fractures.

Currently, the debate on treatment options for mandibular head fractures seems to tip in favor of ORIF [12,26,50,51]. Nowadays, the best osteosynthesis material remains to be chosen. The authors believe that magnesium alloy material is worth considering.

The weaknesses of this study are the relatively small patient group and the limited follow-up period. The strong points of the study are as follows: It is the largest group of patients treated with magnesium osteosynthesis in the mandibular head described thus far (six patients have been described in two publications thus far [30,52]), it is based on the longest observation period of these patients in maxillofacial surgery (a 12-month observation has been described thus far [53]), and it includes a description of changes in the height of mandibular rami, showing the possibility of using multiscrew osteosynthesis (e.g., three low-profile screws per mandibular head, Figure 1, contrary to previously published applications of a single magnesium screw per mandibular head and in one case two screws). This study also includes a reference group and presentation of the results of treatment with solid screws (thus far, only cannulated screws have been described [30]).

## 5. Conclusions

This study shows the clinical advantages of using magnesium resorbable screws for the fixation of fractures of the mandibular heads, obviously achieving union of the bone fragments and restoring the range of contralateral movement in relation to the fractured head and clinically insignificant complications. Magnesium fixation seems to provide at least the same results as titanium fixation but can be expected to disappear over time. It can also be noted that the currently available magnesium screw profiles offer the possibility of multiscrew fixation. Magnesium is a new material in the field of mandibular head osteosynthesis, and hence clinical follow up is recommended to strengthen preexisting evidence.

## Figures and Tables

**Figure 1 materials-15-00711-f001:**
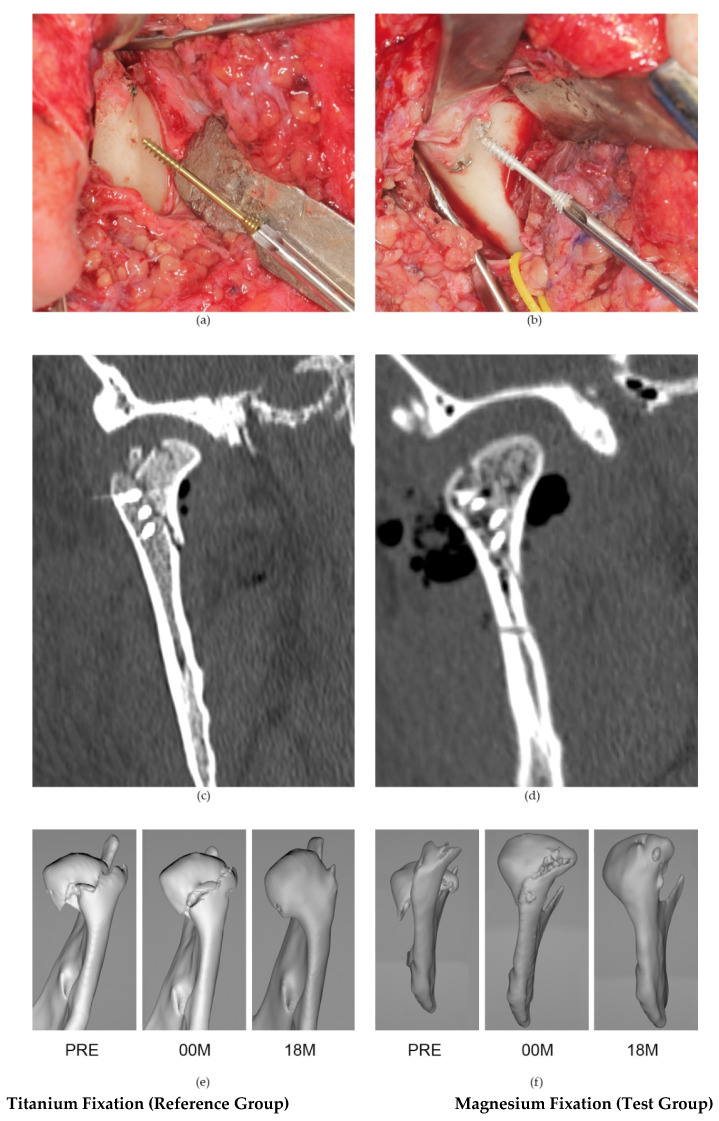
Examples of the clinical use of the 18 mm-long, headless compression titanium screw system 1.8 (**a**) and the 14 mm-long, headless compression magnesium screw system 2.3 (**b**) in the fixation of mandibular head fractures with a preauricular approach. Below, coronal computer tomography scans taken immediately after fixation: (**c**) osteosynthesis performed with three titanium screws and (**d**) osteosynthesis performed with four magnesium screws. Two groups for comparison: follow-up of titanium fixation on the left-hand side (reference group) and magnesium fixation on the right-hand side: (**e**,**f**): PRE-preoperative computer tomography (CT), 00 M—immediate postoperative CT, 18 M—18-month postoperative follow-up.

**Figure 2 materials-15-00711-f002:**
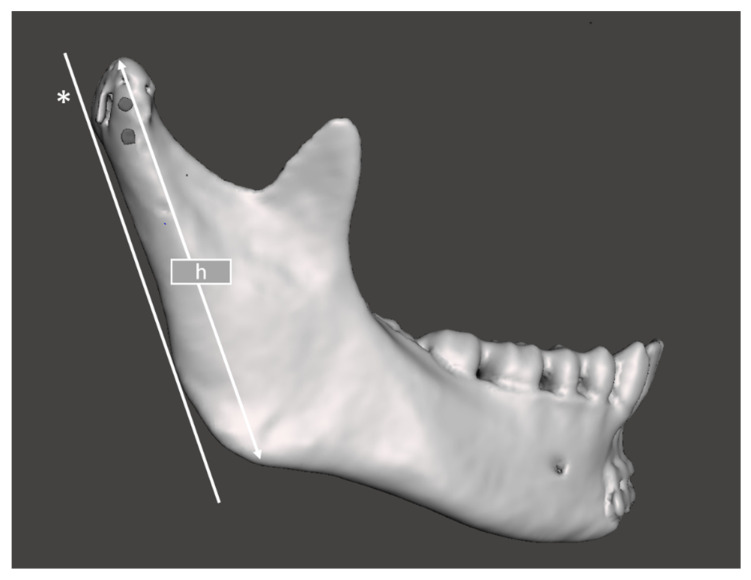
Method of determining mandibular ramus height (h). *—Tangent line to the posterior border of the mandibular ramus. h is determined by the length between the highest and lowest points of the ramus parallel to the tangent line.

**Figure 3 materials-15-00711-f003:**
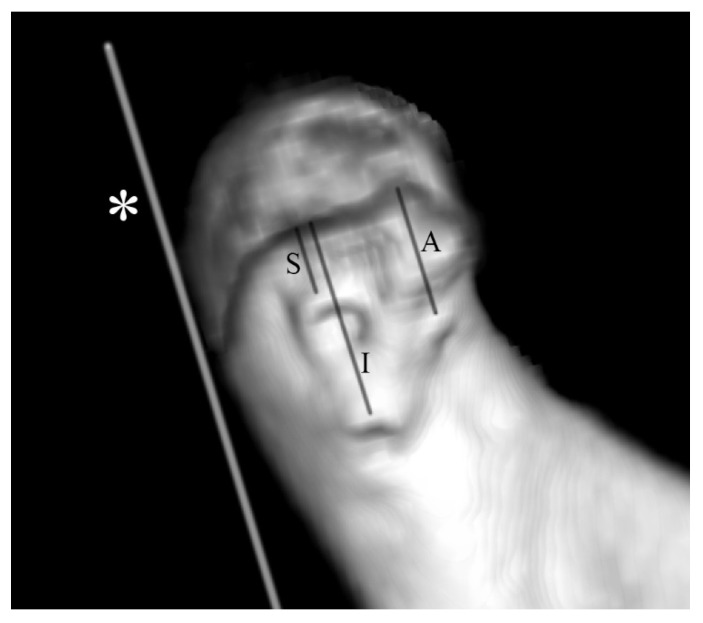
Method for measuring the distance from the screw edge to the fracture line. Measurements were performed using CT volumetric data immediately after fixation of the bone fragments. Example of an imaging study of a patient with osteosynthesis with three magnesium screws of a type C mandibular head fracture on the right side. An asterisk (*) indicates the line tangent to the posterior border of the mandibular ramus in lateral view. Measurements of the superior (S), inferior (I), and anterior (A) screws were taken parallel to the tangent line *.

**Figure 4 materials-15-00711-f004:**
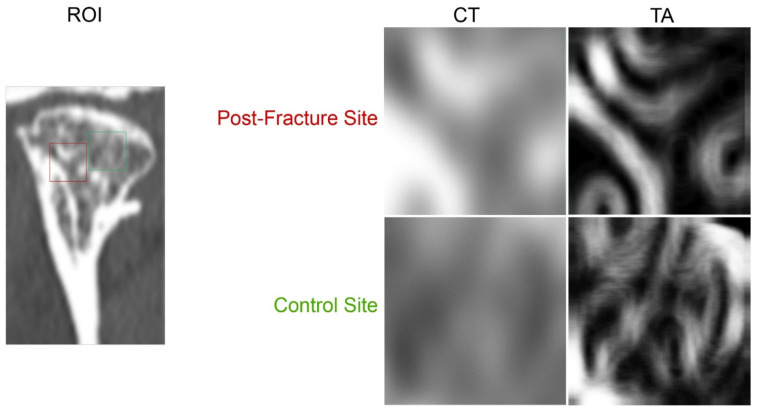
Confirmation of bone union at the fracture fixation site of the mandibular head. Regions of interest (ROIs) are described in the postfracture site in computer tomography (CT) scans taken 18 months after osteosynthesis (red square) and in cancellous control bone (green square). On the right, these areas are presented in an enlarged form (CT). Further to the right, maps of the distribution of the bone texture feature in these ROIs are shown. Texture analysis (TA) includes the sum of the squares of the optical densities acquired by CT in increments of two pixels. A finer pattern of structure can be seen in the control site than in the postfracture site, where there is normal bone interspersed with more opaque bands (*p* < 0.05).

**Figure 5 materials-15-00711-f005:**
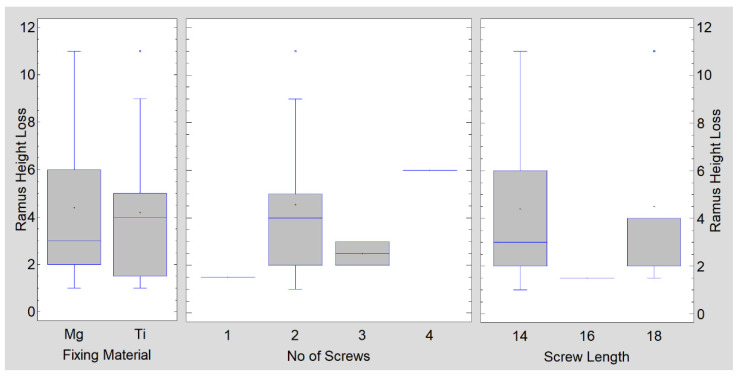
As far as the mandible ramus loss is concerned, the fixing material (*p* = 0.879), number of used screws (*p* = 0.637), and length of used screws (*p* = 0.684) have no statistical significance.

**Figure 6 materials-15-00711-f006:**
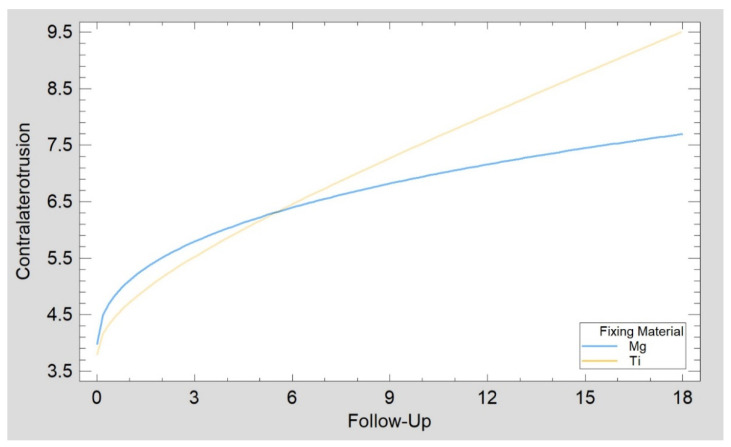
Recovery of contralateral to fracture side range of movement (mm) as a function of time (months) was the same in patients treated with magnesium screws as in the reference group (patients treated with titanium screws). The Kolmogorov–Smirnov statistic of the regression plots (*p* = 0.699) confirmed similar recovery in the test group (Mg) as in the reference group (Ti).

**Figure 7 materials-15-00711-f007:**
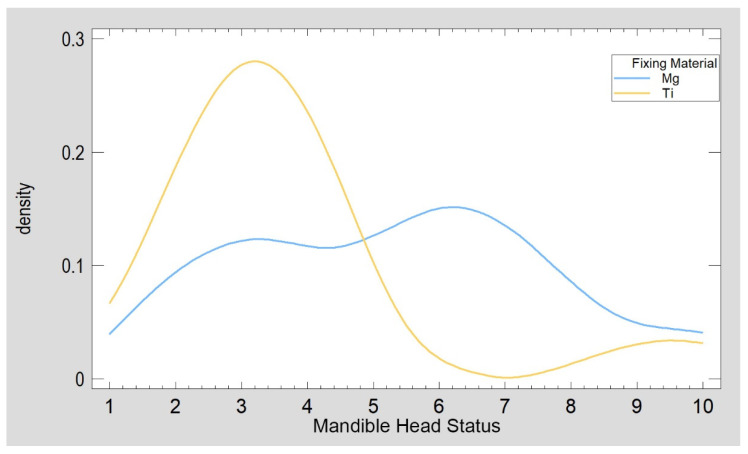
Characteristics of patients included in this study concerning pathomorphological and pathophysiological disturbances after mandibular head injury. The groups were similar (mandible head status, *p* = 0.050), although there were many comminuted fractures in the test (Mg) group.

**Figure 8 materials-15-00711-f008:**
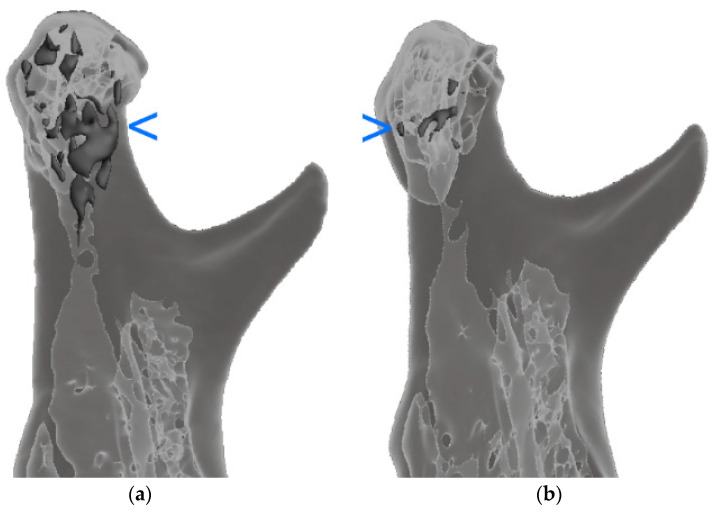
The location of the formed gas is marked in the darkest gray color. Hydrogen volume significantly decreased (*p* < 0.05) during the 3-month observation, i.e., between the third (**a**) and sixth postoperative months (**b**). The uniformly gray areas are an image of compact bone, while the light gray areas are an image of trabecular bone. The blue arrow marks the areas of gas inside the bone. These areas are dark gray.

**Figure 9 materials-15-00711-f009:**
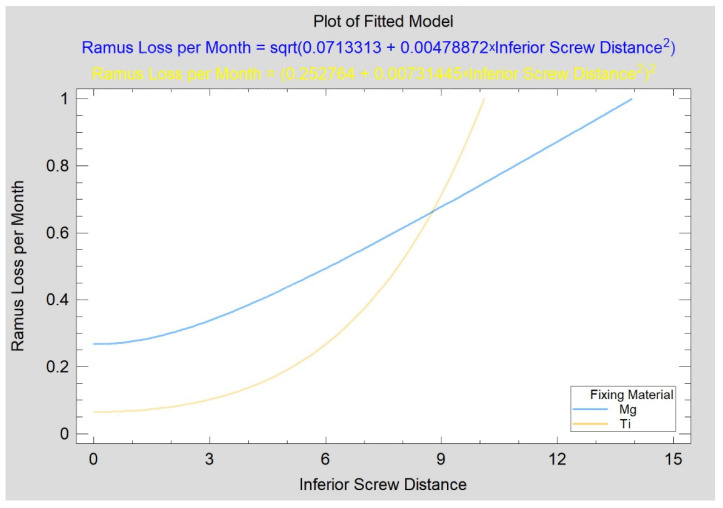
Dependence of mandibular ramus height loss on the position of the inferior screw used for mandibular head fracture treatment. Both variables are given in millimeters. An increasingly lower position of the inferior screw (see “I” distance measurement in Figure 3) from the fracture line was related to an increasing rate of mandibular ramus height loss in the test group (*p* < 0.05), but no such relationship existed in the reference group (*p* = 0.196).

**Table 1 materials-15-00711-t001:** Comparison of the fracture site after healing by means of simple densitometry in computer tomography (CT) and texture analysis (TA).

Measured Feature and Site (i.e., ROI)	Test Group	Reference Group	Between-Group Difference
Optical density at the fracture site	448 ± 122 HU ^1^	408 ± 122 HU ^1^	*p* = 0.453
Optical density in cancellous control bone	419 ± 81 HU ^1^	384 ± 85 HU ^1^	*p* = 0.338
Sum of squares at the fracture site	110 ± 4 ^2^	108 ± 6 ^2^	*p* = 0.486
Sum of squares in cancellous control bone	97 ± 6	94 ± 9	*p* = 0.304

^1^ Hounsfield units; ^2^ the observed opaque islands in the healed fracture site texture generated an increase in TA value.

**Table 2 materials-15-00711-t002:** Comparison of patients treated for mandibular head fractures with magnesium screw (test group) and titanium screw (reference group) fixation.

Variable	Test Group	Reference Group	Between-Group Difference
Age	33 ± 22 years old	38 ± 15 years old	*p* = 0.566
Internal Comorbidity	3 patients	5 patients	*p* = 0.440
Type C Head Fracture	7 patients	7 patients	*p* = 1.000
Type P Fracture	11 patients	8 patients	*p* = 0.415
Dislocation in Joint	9 patients	11 patients	*p* = 0.283
Ramus Shortening	6.1 ± 4.5 mm	8.4 ± 4.5 mm	*p* = 0.261
Head Fragmentation ^1^	3 ± 1	2 ± 1	*p* = 0.175
Occlusion	3 cross, 2 open	7 cross, 4 open	*p* < 0.05
Tooth Injury	4 patients	1 patient	*p* = 0.251
Additional Mandible Fracture	3, body	7, body	*p* = 0.270
Additional Maxillofacial Fracture	Yes (6 patients)	No	*p* < 0.05
Additional Body Injury	Yes (2 patients)	Yes (2 patients)	*p* = 0.918
Delay to Treatment	4.9 ± 3.8 day	5.1 ± 3.8 day	*p* = 0.908
Surgery Duration	230 ± 103 min	258 ± 104 min	*p* = 0.541
Number of Used Screws	2.4 ± 0.7	1.9 ± 0.3	*p* < 0.05
Length of Screws	14 ± 0 mm	16 ± 2 mm	*p* < 0.05
Restored Ramus Height (h)	64.9 ± 5.1 mm	65.4 ± 2.0 mm	*p* = 0.786
Mandible Head Status (factor)	5.5 ± 2.5	3.3 ± 2.3	*p* = 0.052
Helkimo Index Early	4.1 ± 3.0	2.4 ± 2.7	*p* = 0.118
Helkimo Index Late	1.7 ± 2.0	2.4 ± 3.1	*p* = 0.971
03 M Cutaneous Tactile Perception	No Recovery (8)	No Recovery (8)	*p* = 1.000
06 M Cutaneous Tactile Perception	No Recovery (7)	No Recovery (7)	*p* = 1.000
18 M Cutaneous Tactile Perception	No Recovery (3)	No Recovery (1)	*p* = 0.508
00 M Facial Nerve Disfunction	2.7 ± 0.8	2.7 ± 0.6	*p* = 0.933
18 M Facial Nerve Disfunction	1.1 ± 0.3	1.2 ± 0.6	*p* = 1.000
00 M Occlusion	Normal (9)	Normal (6)	*p* = 0.056
00 M MIO	21 ± 4.9 mm	24 ± 10.0 mm	*p* = 0.190
00 M Ipsilaterotrusion	6.0 ± 3.9 mm	4.1 ± 2.7 mm	*p* = 0.259
00 M Contralaterotrusion	3.9 ± 2.2 mm	3.6 ± 1.7 mm	*p* = 0.736
03 M Occlusion	Normal (9)	Normal (8)	*p* = 0.280
03 M MIO	38 ± 5.2 mm	32 ± 7.4 mm	*p* < 0.05
03 M Ipsilaterotrusion	7.7 ± 3.5 mm	5.8 ± 3.9 mm	*p* = 0.323
03 M Contralaterotrusion	6.3 ± 2.8 mm	4.7 ± 2.9 mm	*p* = 0.278
06 M Occlusion	Normal (9)	Normal (7)	*p* = 0.143
06 M MIO	38 ± 10.7 mm	40 ± 4.1 mm	*p* = 0.674
06 M Ipsilaterotrusion	7.2 ± 4.4 mm	10.5 ± 2.5 mm	*p* = 0.196
06 M Contralaterotrusion	5.9 ± 4.5 mm	9.5 ± 1.9 mm	*p* = 0.161
18 M Occlusion	Normal (10)	Normal (8)	*p* = 0.083
18 M MIO	43 ± 3.4 mm	39 ± 3.4 mm	*p* < 0.05
18 M Ipsilaterotrusion	8.3 ± 3.2 mm	10 ± 2.3 mm	*p* = 0.223
18 M Contralaterotrusion	7.8 ± 2.3 mm	8.1 ± 3.0 mm	*p* = 0.870
18 M Ramus Height Loss	4.4 ± 3.3 mm	4.2 ± 3.1 mm	*p* = 0.879
Ramus Height Loss per Month	0.4 ± 0.3 mm	0.7 ± 1.0 mm	*p* = 0.777
18 M Relative Loss of Ramus	7% ± 5%	6% ± 5%	*p* = 0.835

^1^ Number of bone fragments except main distal fragment connected to the ramus and remaining mandible; MIO—maximal interincisal opening; 00 M—immediate postoperational; 03, 06, 18 M—3, 6, 18 months post-operationally.

## Data Availability

The data presented in this study are available on request from the corresponding author. The data are not publicly available due to an ongoing multicenter project.

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
