# Peer review of "Clinical Evaluation of Magnesium Alloy Osteosynthesis in the Mandibular Head"

_materials, 2022, doi:10.3390/ma15030711_

Round 1
Reviewer 1 Report
Well written and well presented. Some suggestions are
1) Explain resorbable materials particularly Mg.
2) The production of Hydrogen gas with magnesium has been described yet not fully addressed. Could you perhaps consider listing the various reactions that can produce hydrogen and why you consider this to be a minor issue?
3) Lines 405-406. Perhaps this can be rewritten in a bit more scientific language.
Author Response
The reviewer guides have been introduced

Reviewer 2 Report
The aim of the paper was to compare open osteosynthesis of mandibular head fractures done either by utilizing resorbable Mg screws or non-resorbable Titanium screws with respect to post-operative outcome (clincal and CT-graphical). The authors found no clinically relevant drawbacks of resorbable Mg screws, thus highlighting the advantages of complete resorption and circumvention of metal removal. In summary, the article lacks a proper structure and does not clearly address the outlined research question within the topic.
Many statements are not placed within the correct corresponding paragraphs – e.g. mixing of results/discussion/introduction. A lot of information is listed within the results section, often also in Figures and tables, a lot of the information being superflous (e.g. elaboration on existing clinical classifications) and not clearly related to the topic of the paper. Figure captions are not well explained. The focus on mandible height loss without comparative relation to the screws utilized is very confusing. Generally speaking, it seems crucial to add an estimation of actual radiographic/CT-graphic bone consolidation, as this is a crucial part of determination of surgical success! Thus, the article is missing a paragraph of actual fracture consolidation assessment. I would recommend adding this. The envisioned topic of osteosythetic material comparison is of high clinical interest and an emerging field of resaerch. Also, the number of reported clinical cases is very promising. I would recommend restructuring the article and clearly adressing the research question. Crucial information with regards to materials and methods is lacking: e.g. was the postoperative asessment done in a blinded fashion? What statistical methods were exactly utilized for which of the asessments? Omitted information on deliberate placement of more screws for Mg material etc.. Thus I would recommend majorly revising the article, in particular the materials and methods section, as well as the figures and captions.
General remarks:
Please try to bring repetitive information into tables to improve the oversight im the article.
Please clearly(!) mark the origin of the material, which following reference 23 seems to be Mg-Y-RE-(Zr) alloy WE43MEO from Meotec GmbH.
Line 31 - It would be helpful for the reader to include the drawbacks of persisting osteosynthetic material in the introduction.
Line 38 – How is the clinical view on the debate of metal removal? Is would make sense to add this information, as this is a clinical article.
Line 76 – To list the classification of nerve damage seems superfluous. The classification can be looked up in literature. Additionally, extended damage with higher scores seems self-explanatory. It would rather be interesting to report if the clinical examination was done by the same person, two independent examiners, in a blinded fashion with respect to the type of screw used and so on.
Line 89 – The same remark as above applies to the Helkimo dysfunction index.
Line 125 – The paragraph on factor analysis is not clear.
Line 139 – According to the table the Mg screws utilized were predominantely shorter that titanium screws. Also, it seems as if more screws were needed for fixation on behalf of the Mg material. Why was this the case? Was this a random occurrence? This needs further explanation. With such a small number of cases treated matched comparisons with regards to fracture type, additional facial fractures and body dimension (height, gender etc.) would have been helpful. As of now, it is not clear wether the differences with repect to screw length and number of screws chosen were due to the material, as e.g. more screws needed as Mg was less rigid or due to chance. The differences in screws used with respect to both material groups thus makes it hard to determine the actual post-op differences related to material properties only.
Line 151 – It was already statetd that no influence of various factors on ramus height loss were identified. It seems superfluous to list every single measurement/comparison of height loss, especially when it was considered to be non-significant.
Line 175 – The topic of the article implies that the aim was to compare fixation screws of two different materials with respect to the surgical outcome. Why was e.g. bone fragment dislocation with relation to ramus height loss examined? This, first of all, is evident and second, does not lie within the scope of the article. This applies for almost the entire paragraph. This is highly confusing.
Figure 4 – Again the relevance with regards to the scope of the article seems unclear. In the caption it is stated, that that the curve is strongly flattened for patients with fractures of more than 3 mm. But according to the graph plot, there was no ramus height decrease larger than 3 mm. Thus, the entire sense of the figure remains unclear.
Line 219 – Again, this paragraph including Fig. 6 seems out of scope.
Linie 276 - The discussion in the end is not intended to elaborate on already known drawbacks of permanent metallic remnants. It is intended to critically discuss results of the study with respect to the intended topic/research question.
Line 292 – To the readers knowledge, in this particular study, pull-out strenghts were not compared! If this was intended as a general remark, this must be clearly stated, as the statement is very misleading! In addition, if this were a result of the study, it would have to be placed in the results section.
Linie 296 – What is meant by torsional properties? This again was not investigated in the study. If this is a general remark, it must be marked as one and also cited in the reference list!
Linie 298 – It is not clear why Mg should be favored with regards to multiscrew fixations. It is argued that due to the material properties, which become weaker over time, osteosynthesis should be performed with more screws. Thus the statement should be, that the use of Mg requires more screws and not vice versa. This is misleading. Aditionally, this remark indicates that this information was known beforehand and was thus intentionally considered while conducting osteosynthesis. Thus a massive misconception is created by marking the comparison of number of screws used for ach material a significant difference, as the number of screws was deliberately chosen. This is biased information.
Line 362 – This is not the scope of the article.
Author Response
The reviewer guides have been introduced.

Reviewer 3 Report
This is a very interesting study with clinical relevance in the use of magnesium vs titanium screws for fixation in mandible condylar head. However, the manuscript must be improved:
There are several doubts about whether Zr of the magnesium alloy is completely resorbed. Include references that support the complete resorption of magnesium alloys. Also, include more references that support the use of magnesium screws.
All the surgeries were performed with the same brand of fixation screws? This information is unclear in the manuscript.
Discuss the real relevance and why the authors support that gas formation during healing is an important issue.
Include the limitations of the study and if the authors recommend further studies.
Consider the results to write the conclusion. The current conclusion seems to be more discussion than the conclusion
Author Response

(The authors gave the same response as above.)

Reviewer 4 Report
The authors present an original research on the use of magnesium alloys for fixation of condylar head after fractures. Although the topic is of interest, the study has major limitations mainly from a methodological point of view, which makes it not suitable for publication on Materials:
- The major limitation is actually the corrosion effect of magnesium alloys. It is very well-documented that magensium alloys tend to corrode. This is a major aspect that has not been evaluated in this study, and although magnesium alloys showed better stability than titanium alloys, magensium alloys might have a more profound side effect due to corrosion.
- The authors mention in the manuscript that this study shows long-term results, after 18 months. The reviewer disagrees that 18 months is a long-term follow up, but is rather a short-term follow up.
- The title could have been better formulated to indicated the type of clinical study (randomised clinical trial? case series? case control? ...... etc)
- The authors do not mention in the materials and methods how they assigned titanium or magensium to the study participants, which is a major flaw of the study design.
- What about the inclusion criteria of patients, in regards to systemic diseases, smoking, systemic drugs.... etc? This is a very important aspect when including patients, but the authors do not mention anything about this.
- What about the calculation of study power and the sample size of the study?
- What is the software used for statistical analysis? What is the name of the statistical test used?
- Control versus test group should be used in text, but the authors have not used these terms.
- In the statistical table, numbers should replace (No).
- The problem of magensium corrosion should be mentioned in the discussion.
- Conclusions are extremely very short.
- English language needs to be revised by a native speaker.
Author Response

(The authors gave the same response as above.)

Round 2
Reviewer 2 Report
Dear authors,
thank you for your comprehensive revision. The manuscript now shows more clarity, better insight and scientific soundness. In light of the major alterations I would recommend to publish the article after considering the following minor adoptions (suggested changes in brackets):
- l. 139/140: Hanging indent
- l. 236/237: In the answer to the reviewers you included box plots which I consider very informative and nice. Please consider to include the three box plots (including statistical conclusion not to be significant) either as Figure or adding it as informative box to preexisting depictions.
- l. 244/245: The screw did not [fully/completey] resorb during...
- l. 255f: Mandible Head Status = 0.0715823 [x] Ramus Shortnening + ...
- l. 278: 3rd -> [third]
- l. 11/90/348: Thanks for your explanation on the specific material used. Though the manufacturer is clearly stated, it still seems unclear which magnesium alloy class was used in particular. This seems important to understand the observed effects. Following your last publications(s) ("Change in pull-out force during Resorption of magnesium compression... / Materials 2021) the specific material class was WE43MEO. If so, please refer accordingly as previously (l. 11/90/348). Otherwise, please give the used composition by chemical measurement (e.g. ICP).
- l. 533: I would recommend to use "disappears [over time]" as the dissolution is no spontaneaous reaction at all
- l. 535/536: I would recommend to delete or phrase more general (e.g. "Magnesium is a new material in the field of mandibular head osteosynthesis and hence clinical follow up is recommended to strengthen preexisting evidence.") since the need for multicenter studies is usually derived from biostatistical evaluation which has not been facilitated in this study.
Author Response
Please, fine the answer in attached file.

Reviewer 3 Report
The requested corrections improved the manuscript
Author Response
Please, fine the answer in attached file as the answer for Reviewer 2 and Reviewer 3.

Reviewer 4 Report
The authors have made an improvement of the manuscript from the previous version. However, the reviewer does not think that the comments raised were endorsed neither in a satisfactory way nor according to the way a reviewer comments should be addressed. First, the reviewer does not understand why the authors respond to the first comment in polish. Second, the reviewer was making a suggestion about the type of study to be included in the title, but the reviewer finds the authors reply unacceptable. The authors should have been better at addressing the comments to the reviewer. Based on that, the reviewer recommends this paper for rejection.
Author Response

(The authors gave the same response as above.)
